# Beyond Pain Management: Skin-to-Skin Contact as a Humanization Strategy in Cesarean Delivery: A Randomized Controlled Trial

**DOI:** 10.3390/healthcare13151866

**Published:** 2025-07-30

**Authors:** José Miguel Pérez-Jiménez, Rocío de-Diego-Cordero, Álvaro Borrallo-Riego, Manuel Luque-Oliveros, Domingo de-Pedro-Jimenez, Manuel Coheña-Jimenez, Patricia Bonilla Sierra, María Dolores Guerra-Martín

**Affiliations:** 1Department of Nursing, Faculty of Nursing, Physiotherapy and Podiatry, University of Seville, c/Avenzoar 6, 41009 Seville, Spain; jpjimenez@us.es (J.M.P.-J.); rdediego2@us.es (R.d.-D.-C.); guema@us.es (M.D.G.-M.); 2Anaesthesiology and Resuscitation Clinical Management Unit, University Hospital Virgen Macarena, 41009 Sevilla, Spain; 3Research Group CTS1149: Integral and Sustainable Health: Bio-Psycho-Social, Cultural and Spiritual Approach for Human Development, 41009 Seville, Spain; 4Research Group CTS-1158: Development and Evaluation of Interventions in Health and Education, 41009 Seville, Spain; 5Indorama Ventures Quimica, S.L.U., Pol. Ind. Guadarranque, sn., 11360 San Roque, Spain; dodepeji@gmail.com; 6Podiatry Department, Faculty of Nursing, Physiotherapy and Podiatry, University of Seville, 41009 Seville, Spain; mcohena@us.es; 7Research Group CTS589: Advances in Podiatric Surgery, 41009 Seville, Spain; 8Department of Health Sciences, Technical Private University of Loja, Loja 110107, Ecuador; pbonilla65@utpl.edu.ec

**Keywords:** skin–to–skin contact, cesarean section, postoperative pain, uterine contraction, maternal satisfaction, humanized care

## Abstract

Background: Postoperative pain management after a cesarean section remains a significant challenge, as inadequate control can delay maternal recovery and hinder early bonding and breastfeeding. While multimodal analgesia is the standard approach, non–pharmacological strategies like immediate skin–to–skin contact (SSC) are often underused despite their potential benefits in reducing pain, improving uterine contractions, and increasing maternal satisfaction. Objective: To evaluate the effects of immediate SSC on postoperative pain perception, uterine contraction quality, and maternal satisfaction, and to explore ways to incorporate SSC into routine post–cesarean care to promote recovery and humanized care. Method: A randomized clinical trial was conducted with 80 women undergoing elective cesarean sections, divided into two groups: SSC (40 women) and control (40 women). Postoperative pain was measured using the Visual Analog Scale (VAS) at various intervals, while uterine contraction quality and maternal satisfaction were assessed through clinical observation and a Likert scale, respectively. Results: We found that women in the SSC group experienced significantly lower pain scores (VAS2 and VAS3, *p* < 0.001), stronger infraumbilical uterine contractions (92.5%, *p* < 0.001), and higher satisfaction levels (average 9.98 vs. 6.50, *p* < 0.001). An inverse correlation was observed between pain intensity and satisfaction, indicating that SSC enhances both physiological and psychological recovery. Conclusions: Immediate SSC after cesarean is an effective, humanizing intervention that reduces pain, supports uterine contractions, and boosts maternal satisfaction. These findings advocate for integrating SSC into standard postoperative care, aligning with ethical principles of beneficence and autonomy. Further research with larger samples is necessary to confirm these benefits and facilitate widespread adoption in maternity protocols.

## 1. Introduction

Cesarean sectioning is a surgical procedure that is often necessary to ensure the health of the mother and child. However, it can involve several complications and challenges in recovery. One of the most common problems faced by mothers after cesarean section is postoperative pain, which can be intense and prolonged and affect their physical and emotional well–being [1].

Postoperative pain following a cesarean section affects up to 80% of patients within the first 24 h, prolonging recovery, increasing maternal morbidity, and impairing early mother–infant bonding [2]. Excessive opioid use may lead to adverse effects for both mother and infant, making multimodal analgesia essential to optimize pain control while minimizing medication exposure [3].

Skin–to–skin contact reduces pain perception, decreases stress, and promotes physiological stability for both mother and newborn [4]. Since adequate pain management is a fundamental human right [5], integrating skin–to–skin contact into post–cesarean care protocols can accelerate recovery and enhance the maternal–infant experience [4,6].

In this context, skin–to–skin contact (SSC) is a practice that not only promotes bonding between the mother and child but can also have a positive impact on the mother’s recovery after surgery [7].

Skin–to–skin contact (SSC) after a cesarean section is not only an evidence–based practice but should also be recognized as a fundamental right of mothers and newborns [6]. According to international guidelines on the right to a humanized birth, such as the recommendations of the World Health Organization (WHO) [8] and the Baby–Friendly Hospital Initiative (BFHI) [9], all women should have the opportunity to establish immediate contact with their babies, regardless of the mode of delivery. Implementing SSC ensures respect for maternal dignity, promotes autonomy, and strengthens early bonding with the newborn.

Moreover, SSC helps reduce health inequality by addressing the persistent gap in post–cesarean care quality. In many healthcare systems, routine mother–infant separation remains a common practice, negatively impacting breastfeeding, maternal emotional stability, and neonatal well–being [10]. The standardized adoption of SSC not only improves clinical and emotional outcomes for both mothers and babies but also fosters a more equitable and humanized model of care [11], ensuring that all mothers and newborns receive high–quality attention, regardless of their socioeconomic background or the healthcare facility where they are attended.

Skin–to–skin contact involves placing the newborn on the mother’s bare chest immediately after birth, which facilitates thermal regulation, breastfeeding, and emotional bonding. Recent research has shown that SSC can have beneficial effects in reducing pain and anxiety in mothers who have had cesarean sections.

For example, a study by Moore et al. [10] found that mothers who experienced SSC reported lower levels of pain and better satisfaction with their birth experience compared to those who did not. In addition, the release of hormones such as oxytocin during SSC may contribute to decreased pain, improved emotional state of the mother, and secure attachment to her child [12]. Most studies have found that mothers report that the pain they feel in the postoperative period after cesarean section prevents them from holding and breastfeeding their babies, and recommend further research on this aspect in randomized clinical trials [13,14].

SSC in the immediate postoperative period of a cesarean section not only aligns with a humanized care approach but is also grounded in the bioethical principles of beneficence and non–maleficence. Scientific evidence shows that SSC enhances maternal–infant well–being by promoting neonatal thermal stability, reducing stress and pain, and strengthening the mother–child bond. Additionally, by minimizing early separation, SSC helps prevent complications, such as neonatal hypoglycemia, disruptions in the initiation of breastfeeding, and increased maternal stress, which can negatively impact postoperative recovery and neonatal adaptation [15]. Implementing SSC safely in the context of a cesarean section not only optimizes clinical outcomes but also ensures respectful and dignified care for both the mother and newborn, reducing the unnecessary risks associated with conventional practices that promote early separation.

Despite these benefits, the implementation of skin–to–skin contact in cesarean deliveries is often hampered by several factors, such as lack of support from medical staff, postoperative maternal conditions, and hospital policies. This highlights the need for increased awareness and training regarding the importance of SSC in the context of cesarean section [16].

As we seek to improve the birth experience and recovery of mothers, it is essential to consider how simple techniques, such as SSC, can transform the recovery process and strengthen the mother–child bond, thus contributing to a more positive and healthy birth experience.

Nursing teams are in direct contact with patients and are part of the first line of care at all levels of healthcare, with special attention to pain management [17].

Considering the available evidence and the context of the cesarean surgical procedure, limited information is available regarding the connection between immediate skin–to–skin contact during cesarean sections and its impact on uterine contractions and postpartum pain. Therefore, our main objective is to investigate whether immediate skin–to–skin contact between a healthy newborn and the mother after cesarean section promotes uterine contractions when leaving the post–anesthesia recovery zone and helps reduce postoperative pain. In addition, we will examine how this technique relates to other factors, such as maternal satisfaction. This study aims to evaluate the impact of immediate skin–to–skin contact (SSC) on postoperative pain perception in women undergoing cesarean section, assessing its relationship with the quality of uterine contractions in the postoperative period. Additionally, it seeks to analyze maternal satisfaction with the SSC experience compared with conventional care, identifying its benefits for recovery and mother–child bonding. Based on the findings, this study proposes strategies for implementing SSC in the context of cesarean delivery, promoting a humanized and safe approach to ensure its adoption as a standard of maternal–neonatal care.

## 2. Materials and Methods

In accordance with CONSORT guidelines for clinical trials [18], we designed a randomized, unblinded study enrolling women undergoing cesarean section in the Gynaecology and Obstetrics Department. Ethical approval was granted by the hospital’s Ethics, Clinical Research and Bioethics Committees (see Appendix A). The trial adhered to the Declaration of Helsinki and complied with Organic Law 3/2018 of 5 December on Personal Data Protection and Digital Rights. All participants received written information and provided written informed consent. Trial identification number: UTN: U1111-1238-8710; RBR-67gq6k; https://ensaiosclinicos.gov.br/rg/RBR-67gq6k/ (accessed on 22 October 2024).

### 2.1. Eligibility Criteria:

Inclusion criteria: Pregnant women aged between 20 and 40 years, with fetal positions that are not suitable for vaginal delivery (such as breech), cephalopelvic disproportion that would hinder vaginal delivery, or previous placental issues that prevent vaginal delivery. This also includes cases in which cesarean sections need to be repeated due to the continuation of the previous indication or the emergence of a new indication that differs from the one that led to the previous surgery. Women whose newborns are immediately evaluated by a neonatologist and deemed healthy, with an Apgar score of 9 or higher in 5 min. Cesarean delivery is performed between 37 and 41 weeks of gestation. Pregnancies and immediate postpartum periods must be free of any fetal abnormalities or malformations, regardless of their nature or cause, and this information should be documented in the medical records.

Exclusion criteria: Women whose cesarean section has been complicated by a significant medical condition (such as cardiac, metabolic, or respiratory diseases). Emergency cesarean sections due to situations like umbilical cord prolapse, premature separation of the placenta, previous placental issues with heavy bleeding, suspected fetal distress, or suspected uterine rupture.

### 2.2. Randomization and Blinding Methods

The sample selection was performed using simple random sampling. Patients diagnosed with the need for a cesarean section were recruited for the clinical trial during high–risk consultation for pre–evaluation. The members of the research team provided information about this study and invited them to participate. Those who met the inclusion criteria, were adequately informed, voluntarily provided consent, and signed the consent form were included in this study (see Appendix A). In the Research Unit, participants were anonymized by assigning an order number to their medical records, which served as their identification. Participants were allocated through permuted randomization using blocks of size 4. The pregnant women were informed of the randomization results upon entering the operating room.

In compliance with the principles of autonomy, beneficence, and non–maleficence, we secured informed consent from all participants. Each woman was given a clear, comprehensive account of this study’s objectives, procedures, and potential benefits and risks in an environment free from pressure, with ample time for reflection. They were encouraged to ask questions at any point and reminded of their right to withdraw without any impact on their medical care. A humanized approach was adopted, using plain language to ensure full understanding. To safeguard well–being and decision–making capacity, women unable to provide consent due to medical incapacity were excluded.

### 2.3. Intervention

Group A (Group no SSC): This group followed the standard intervention in accordance with the hospital’s protocols. After the cesarean section, the mother was moved to the post–anesthesia recovery room, while the newborn was taken in a crib to the maternity ward with the father or a companion. Group B (Intervention Group—SSC): In this group, skin–to–skin contact with the newborn was initiated in the operating room and continued in a designated room within the postpartum unit, with a companion present. The following variables were studied:

Sociodemographic variables: age [years], weight [kg], height [cm], BMI [kg/m^2^], and education level [primary, secondary, or higher education].

Skin–to–skin contact (SSC): when the infant spent at least one hour after the cesarean section on the mother’s torso, in the operating room. Dependent variables: Postoperative pain: pain manifestation in the operating room (VAS1), when the mother arrived in the recovery room (VAS2) and when she left the recovery room, for admission to an obstetrics and gynecology room (VAS3). The VAS (Visual Analog Scale) ranges from 0 to 10, where 0 is no pain and 10 is unbearable pain. Uterine contraction: uterine contraction level prior to the mother’s discharge from the awakening room. The anesthesiologist gently pressed on the mother’s abdomen to palpate the uterus, check its firmness, and assess whether the uterus was very well contracted at the infraumbilical level or if, on the contrary, it remained at the umbilical or supraumbilical levels. Mother’s satisfaction degree—SSC: satisfaction with performance of SSC. A 10–point Likert scale was used: Very dissatisfied, Dissatisfied, Neither satisfied nor dissatisfied, Satisfied, and Very satisfied. Newborn Apgar test at 1 and 5 min: expressed in whole numbers. Rapid test at birth by which the pediatrician evaluated Appearance, Pulse, Irritability, Activity, and Respiration.

### 2.4. Statistical Analysis and Sample Size Calculation

The sample size was calculated to detect a 1.5–point difference on the VAS (0–10 scale), assuming a standard deviation (SD) of 1.9 based on Crenshaw et al.‘s study [19]. With a significance level of 0.05 and 90% power, 34 participants per group were required. This was increased to 40 per group to account for potential dropouts (resulting in a large effect size, Cohen’s *d* = 0.79). Post hoc analysis confirmed that even with a moderate effect size (Cohen’s *d* = 0.5), power would remain at 80%. First, a descriptive analysis of all variables was performed. The qualitative variables are presented as absolute and relative frequencies. Quantitative variables are presented as mean and standard deviation. In the univariate inferential analysis, the chi–square test or Fisher’s test and the Student’s *t*-test or Mann–Whitney U-test were used, depending on whether the patient had a normal distribution. The relationship between continuous variables was analyzed using the Spearman correlation. The Kruskal–Wallis test with Bonferroni correction was applied to assess the relationship between the categories of uterine contraction and VAS2 and VAS3. The statistical significance level was set at a *p*-value of 0.05. The statistical analysis was performed using the IBM SPSS 26 and JAMOVI 2.3.28 software programs. The graphs were generated using Microsoft Excel 365, version 2504, build 18730.20168.

## 3. Results

A total of 339 pregnant women who were indicated for a cesarean section were assessed for eligibility, and 23.9% (80 women) were randomized during the study period. They were recruited based on the inclusion criteria and randomly assigned to either the “SSC Experimental Group” (*n* = 40) or the “Control Group” (*n* = 40). Additionally, 2.3% (*n* = 8) chose not to participate in this study (see Appendix A). The descriptive analysis and cesarean section data for the variables by group (Control—SSC) are shown in Table 1. Data were also gathered regarding the cesarean section process, with a focus on both the mother and the newborn. Notably, there were highly significant correlations between SSC and pain levels (VAS2 and VAS3), with lower pain scores observed in the SSC group, suggesting its effectiveness in post–cesarean pain management. Additionally, uterine contractions were more frequent and of better quality in the SSC group, with up to 92.5% occurring at the infraumbilical level, indicating a potential physiological benefit of early maternal–neonatal interaction. Maternal satisfaction was also significantly higher in the SSC group (*p* < 0.001), with an average score of 9.98 compared with 6.50 in the control group, reinforcing the positive impact of SSC on postpartum experience. Furthermore, although the sociodemographic characteristics of both groups were generally homogeneous, analysis revealed a significant difference in education level (*p* < 0.001), and women with lower educational attainment experienced greater postoperative pain intensity, which may have influenced their perception and acceptance of SSC. The correlational analysis between quantitative variables revealed significant associations in the SSC group. Postoperative pain levels measured using VAS3 were significantly negatively correlated with maternal satisfaction (*p* = 0.040), suggesting that lower pain levels are directly linked to higher maternal satisfaction in mothers who practiced skin–to–skin contact. In contrast, no significant correlations were found in the control group between postoperative pain and maternal satisfaction, reinforcing the hypothesis that SSC not only reduces pain but also enhances the post–surgical experience. Additionally, correlations with Apgar scores in both groups were not significant, indicating that SSC implementation did not negatively impact immediate neonatal outcomes. These findings support the integration of SSC into post–cesarean care protocols to optimize both maternal recovery and the overall childbirth experience (Table 2). As shown in Figure 1, the SSC group exhibited a significant inverse correlation between VAS3 scores and maternal satisfaction, such that lower post–cesarean pain intensity was associated with higher satisfaction levels (*p* < 0.05). When plotting pain scores (VAS 0–10) against overall maternal satisfaction (0–10), a steep negative slope emerged, with a correlation coefficient of -0.68 (r^2^ = 0.46; *p* < 0.05), indicating that each one–point decrease in pain corresponded, on average, to nearly a one–point increase in satisfaction. Moreover, the tight clustering of data points around the regression line shows consistency: 80% of women reporting VAS ≤ 3 at six hours after surgery achieved satisfaction scores ≥ 8, whereas only 40% of those with VAS ≥ 5 reached the same satisfaction threshold. Clinically, these findings suggest that SSC not only reduces pain perception but also directly enhances maternal well–being and confidence in neonatal care, factors essential for successful breastfeeding initiation and mother–infant bonding. This finding aligns with previous research indicating that physical closeness and immediate skin–to–skin contact contribute to reduced stress and improved postoperative comfort [4,6]. Table 3 presents the post hoc analysis of the relationship between pain levels (VAS2 and VAS3) and uterine contraction location. The results indicate statistically significant differences, particularly between infraumbilical and supraumbilical contractions (VAS2: 23.73; VAS3: 34.31, *p* < 0.05) and infraumbilical and umbilical contractions (VAS2: 24.1; VAS3: 28.01, *p* < 0.001). These findings suggest that SSC may contribute to more effective uterine contractions, particularly at the infraumbilical level, which could facilitate postpartum recovery. The absence of significant differences between supraumbilical and umbilical contractions highlights the need for further investigation into the physiological mechanisms underlying these effects.

Figure 2 presents the relationship between uterine contractions and VAS3 pain levels, highlighting that the absence of pain is predominantly observed at infraumbilical contraction levels. This finding reinforces the potential benefit of SSC in optimizing uterine contraction efficiency while reducing postoperative pain. The clear inverse association between higher contraction quality and lower pain scores supports the hypothesis that SSC contributes to a more favorable recovery process by promoting physiological responses that enhance maternal comfort and satisfaction. Further studies with larger sample sizes could provide additional insights into the mechanisms underlying this association.

As shown in Figure 3, infraumbilical uterine contraction strength (0–10 scale) demonstrated a moderate inverse correlation with VAS2 pain scores, with a correlation coefficient of -0.52 (*r^2^* = 0.27; *p* = 0.02). This indicates that each one–point increase in contraction intensity corresponded, on average, to a half–point decrease in pain perception. Data clustering around the regression line further highlights consistency: 70% of women reporting contraction scores ≥ 6 experienced VAS2 ≤ 3, whereas only 30% of those with contraction scores ≤ 3 fell below that pain threshold. Clinically, this pattern raises the possibility that stronger uterine activity, and the associated surge in endogenous oxytocin, may exert an intrinsic analgesic effect after cesarean delivery.

## 4. Discussion

Skin–to–skin contact (SSC) has been shown to significantly improve postoperative pain management in cesarean patients by promoting maternal well–being and enhancing physiological stability, including more effective uterine contractions. This non–pharmacological approach aligns with bioethical principles of beneficence and autonomy, as it minimizes the need for excessive analgesic use while empowering mothers during their recovery process. However, to draw more conclusive results, future studies should include larger sample sizes to strengthen the evidence supporting SSC as a standard practice in post–cesarean care, ensuring equitable and humanized pain management.

This technique was also associated with higher levels of mothers’ satisfaction regarding their childbirth experience, especially compared to previous cesarean deliveries. A research team led by Margaret M. Boyd conducted a quality improvement project to establish a standard of care. They found that SSC boosts maternal oxytocin and beta–endorphin levels, promoting uterine contractions, although they did not show a direct effect on pain reduction [20]. However, our study does provide evidence of this. Zhang et al. conducted a multicenter randomized controlled trial in China with 720 women undergoing elective cesarean section to determine the association between the duration of skin–to–skin contact after cesarean delivery and breastfeeding outcomes, as well as its impact on maternal and neonatal health. They found that a longer duration of skin–to–skin contact significantly improved the rate of early breastfeeding initiation and exclusive breastfeeding at discharge. Additionally, it was associated with reduced postpartum blood loss and a lower admission rate to the neonatal intensive care unit [21]. Women who have undergone cesarean section are known to have lower levels of circulating oxytocin, a hormone crucial for bonding and maternal pain control, making immediate SSC even more essential for women who have lower levels. This was supported by Kenkel [22], who emphasized the importance of this technique, although their work does not include primary quantitative data like our study; instead, it is a narrative review.

The analysis of sociodemographic variables, particularly education level, indicates that women with lower educational attainment reported greater postoperative pain, which may have influenced their perception and acceptance of skin–to–skin contact (SSC). Likewise, Hussen et al. [2] observed at the Hawassa University Comprehensive Specialized Hospital that mothers with lower education described higher pain levels following cesarean section. These findings underscore maternal education as a key determinant in the post–cesarean experience and in the adoption of skin–to–skin contact within a humanized, evidence–based approach.

By focusing on pain as the main variable, we found that women who experienced SSC reported lower levels of postoperative pain upon arriving in the recovery room. Some studies have explored this from an intraoperative perspective, like Kollmann et al., whose study was conducted in a tertiary hospital in Austria. However, their results were inconclusive [23]. To date, most published research on SSC has focused on vaginal delivery and other aspects. This study aimed to analyze variables related to women’s mood and comfort and their connection to SSC. The findings revealed that patient satisfaction and satisfaction levels were very high. These results align with those of other studies, indicating that women who initiated SSC during surgery reported greater satisfaction and lower salivary cortisol levels over time [23]. It is crucial to support mothers during such a traumatic experience as a cesarean section. Regularly asking patients “What do you need?” can help foster a more humanized care environment, reduce their pain, and enhance their comfort [24]. This perspective aligns with recent contributions such as that of Díaz–Ogallar et al. [25], published in *Healthcare*, which developed a predictive model to facilitate immediate SSC after childbirth. Although their sample was large, it was a cross–sectional study that did not directly assess postoperative pain or the quality of uterine contractions, which limits its applicability to surgical contexts like ours. Some studies refer to the concept of “proximity pleasure”, which describes the joy a mother feels from being close to her baby [26]. Recent studies, such as that by McCutcheon et al. [27] in the *International Journal of Environmental Research and Public Health*, highlight increased maternal satisfaction associated with SSC during cesarean sections. However, as it is an integrative review, it does not provide direct empirical evidence regarding postoperative pain or physiological parameters such as the quality of uterine contractions.

Similarly, Espinós Ramírez et al. [28], in *Medicina*, compared pharmacological analgesic strategies following cesarean section and found that intrathecal morphine improves pain control. Nevertheless, their study does not address non–pharmacological alternatives such as SSC, nor does it explore SSC’s relationship with maternal satisfaction or emotional recovery—key aspects that are thoroughly examined in our clinical trial.

If maternity hospitals do not facilitate immediate SSC after cesarean sections, many mothers and their newborns may miss out on the potential benefits of this approach [4]. Techniques like cesarean section should not prevent SSC from being performed in the operating room, as this study has shown that it can be performed easily. In summary, skin–to–skin contact among women undergoing cesarean sections enhances uterine contractions, increases maternal satisfaction, and helps reduce maternal pain. Thanks to this research and collaboration with other clinical management units, an SSC protocol has been developed and implemented in this hospital, which also includes emergency cesarean sections.

The implementation of skin–to–skin contact (SSC) after a cesarean section not only enhances the maternal–infant experience but also provides physical, emotional, and spiritual benefits. It helps alleviate postoperative pain, strengthens the mother–infant bond, and promotes a more satisfying recovery. However, its application faces barriers such as inadequate postoperative pain management, lack of staff training, and resistance to modifying conventional practices [6]. Ensuring effective pain control is a fundamental human right [5] and a key factor in maternal well–being because it enables mothers to actively participate in SSC, fostering early bonding and improving their post–surgical experience.

To overcome these challenges, it is essential to train healthcare professionals in multimodal analgesia [29], establish evidence–based protocols, and adapt surgical environments to facilitate the safe implementation of SSC. Additionally, hospitals and maternity units should standardize this practice without compromising maternal–neonatal safety, ensuring proper monitoring and promoting continuous nursing support [6].

Systematically and humanely integrating SSC into clinical practice contributes to more equitable, person–centered care, which is aligned with the ethical principles of beneficence and autonomy. Its incorporation not only enhances the quality of care but also optimizes clinical outcomes and maternal satisfaction. The inconsistency in SSC implementation may contribute to disparities in health outcomes, highlighting the importance of recognizing it as a fundamental right [6].

SSC following a cesarean section not only provides physiological and emotional benefits for the newborn but also strengthens the maternal role, respecting the dignity and autonomy of the mother [4]. This approach aligns with person–centered care models [30], enabling mothers to play an active role in their child’s well–being and foster early bonding and breastfeeding. Nursing plays a pivotal role in the implementation of SSC by offering companionship, emotional support, and reassurance during highly vulnerable moments. Effective communication ensures that care is tailored to each mother, enhancing her confidence and overall well–being, while a supportive environment and continuous training further cultivate compassionate care [31]. However, challenges such as workload pressure and stress may hinder proper implementation. Integrating technical expertise with a humanized approach in nursing practice optimizes the SSC experience, ensuring a safe environment and promoting a more positive and respectful birth experience [32].

### Study Limitations

The principal limitation of this study was the unavailability of the hospital’s post–anesthesia recovery room, which necessitated the adaptation of a maternity suite to enable the mother, her newborn, and the accompanying person to remain together following the cesarean section. Although this logistical workaround may have influenced both the experience and perception of immediate skin–to–skin contact, it nevertheless demonstrates the feasibility of preserving family–centered care in constrained settings.

The sample size was modest, potentially limiting the generalizability of our findings to other clinical environments. However, the results obtained offer valuable preliminary insights into immediate post–cesarean skin–to–skin practices. To strengthen and validate these observations, we plan a follow–up multicenter study with an expanded cohort drawn from hospitals across Spain, thereby enhancing statistical power and external validity.

Finally, the potential for bias due to lack of blinding should be acknowledged. Participants were aware of their group allocation prior to entering theater, as informed consent was required before anesthesia. While this may have shaped expectations, the consistency of responses across groups suggests that the impact on our core outcomes was limited.

## 5. Relevance to Clinical Practice and Impact Statement

It is common practice to separate mothers from their newborns after a cesarean section, which means that there is often no immediate skin–to–skin contact (SSC). This can result in inadequate pain management for the mother. To date, research on early neonatal SSC and maternal pain has been limited. Unfortunately, early SSC has not been widely adopted as a standard practice in most cesarean deliveries, and the reasons for this remain unclear. Procedures like cesarean sections should not hinder skin–to–skin contact, as this can lead to poor pain control.

To ensure the safe and effective implementation of immediate skin–to–skin contact (SSC) after a cesarean section, standardized protocols that guide its application in hospitals and maternity units are essential. Additionally, healthcare staff training is crucial to raise awareness of its impact on postoperative pain reduction, uterine contraction quality, and maternal satisfaction [29]. Optimizing pain management with multimodal analgesia, adapting surgical and postoperative spaces to facilitate SSC, and strengthening follow–up through maternal satisfaction assessments will help consolidate SSC as a standard practice. These strategies promote a humanized and evidence–based approach, are aligned with bioethical principles, and focus on maternal–neonatal well–being.

## 6. Conclusions

Immediate skin–to–skin contact during cesarean sections offers numerous benefits, making it very beneficial not to separate the mother and baby in the initial hours after the procedure. Achieving skin–to–skin contact directly from the operating room is quite feasible. In addition, skin–to–skin contact in women undergoing cesarean sections reduces postoperative pain levels. The contractions of the uterus immediately following a cesarean are more effective when the mother and baby are in skin–to–skin contact, which helps to lessen the sensation of pain. Higher uterine contraction frequency is associated with lower maternal pain. Furthermore, mothers who have skin–to–skin contact during cesarean sections report greater satisfaction and comfort, and these experiences are often more fulfilling compared to previous cesarean deliveries without SSC.

Immediate skin–to–skin contact (SSC) after a cesarean section has been shown to be an effective strategy for reducing postoperative pain, improving the quality of uterine contractions, and enhancing maternal satisfaction. This intervention not only optimizes clinical outcomes but also reinforces a humanized approach to care, aligning with the bioethical principles of beneficence and autonomy. Despite its benefits, the implementation of SSC faces challenges, such as the lack of standardized protocols and resistance to changes in clinical practice. Therefore, it is essential to promote studies with larger sample sizes to further support its efficacy and to raise awareness among healthcare professionals for its adoption as a standard post–cesarean care practice.

## Figures and Tables

**Figure 1 healthcare-13-01866-f001:**
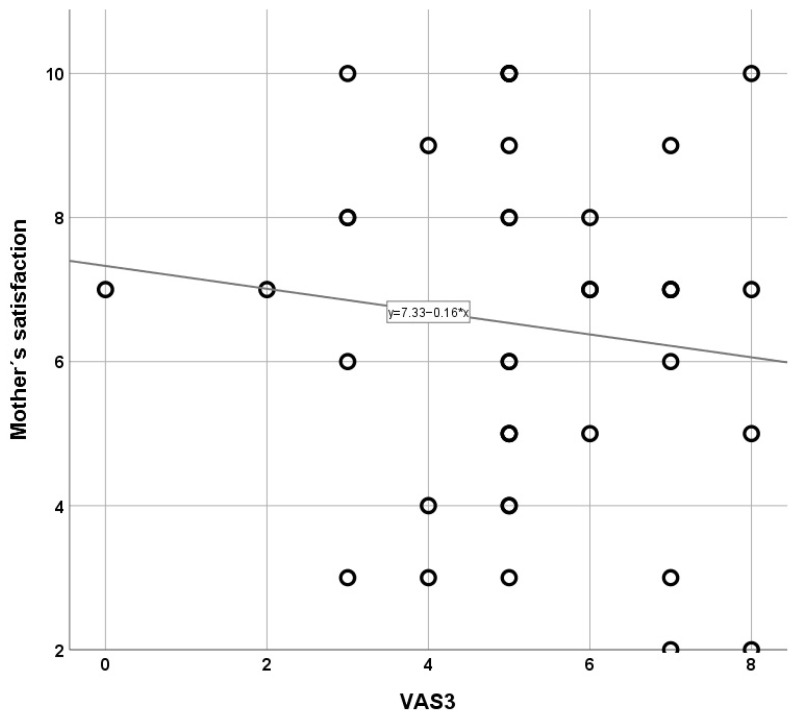
Relationship between mother’s satisfaction and VAS3 in the SSC group.

**Figure 2 healthcare-13-01866-f002:**
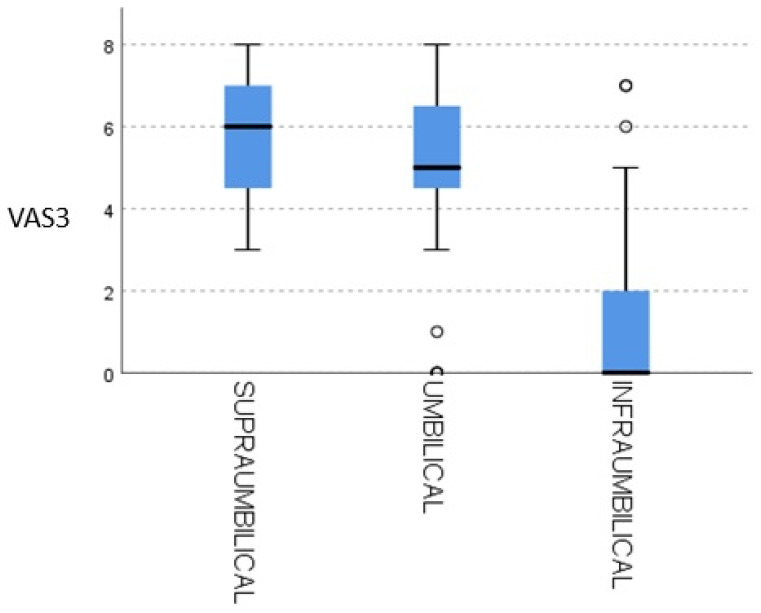
Uterine contraction in relation to VAS3.

**Figure 3 healthcare-13-01866-f003:**
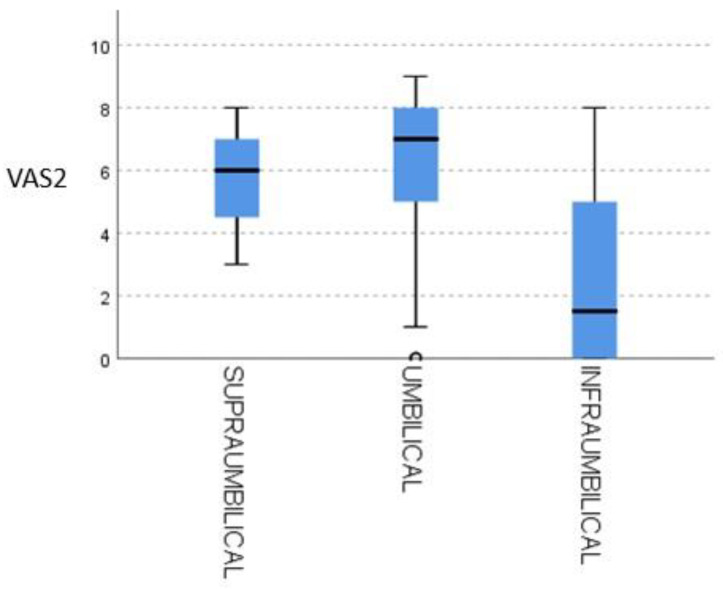
Uterine contraction in relation to VAS2.

**Table 1 healthcare-13-01866-t001:** Sociodemographic data according to groups and cesarean section data according to groups (Control—SSC). *N* = 80.

		Control *(n* = 40)	SSC (*n* = 40)	*p*-Value
Age (years)		33.12 (SD 3.57)	33.20 (SD 3.16)	0.921
Weight (kg)		87.23 (SD 11.16)	86.55 (SD 13.28)	0.840
Height (cm)		166.25 (SD 6.77)	163.25 (SD 5.86)	0.051
BMI (kg/m^2^)		31.70 (SD 4.59)	32.59 (SD 5.50)	0.498
Level of studies	Primary	23 (57.5%)	9 (22.5%)	<0.001
	Secondary	5 (12.5%)	25 (62.5%)	
	Higher	12 (30%)	6 (15%)	
Cesarean Section Data		Control (*n* = 40)	SSC (*n* = 40)	*p*-Value
Pain	VAS1	0.78 (SD 1.90)	0.57 (SD 1.80)	0.890
	VAS2	6.23 (SD 2.12)	1.48 (SD 2.11)	<0.001
	VAS3	5.23 (SD 1.77)	0.60 (SD 1.17)	<0.001
Mother’s satisfaction		6.50 (SD 2.41)	9.98 (SD 0.16)	<0.001
Uterine Contraction	Umbilical	28 (70%)	3 (7.5%)	<0.001
	Infraumb.	9 (22.5%)	37 (92.5%)	
Apgar	0	8.68 (SD 0.76)	9.20 (SD 1.02)	0.004
	1	9.18 (SD 0.60)	9.48 (SD 0.85)	0.008
	5	9.90 (SD 0.30)	9.93 (SD 0.27)	0.694

**Table 2 healthcare-13-01866-t002:** Correlational analysis of pain (VAS2/VAS3) with maternal satisfaction and Apgar scores by group (Control/SSC, *n* = 80).

		Mother’s Satisfaction	APGAR0	APGAR1	APGAR5
Control (*n* = 40)	VAS2	−0.090	−0.180	−0.241	−0.241
	VAS3	−0.107	−0.266	−0.210	−0.104
SSC (*n* = 40)	VAS2	−0.198	−0.033	−0.031	−0.135
	VAS3	−**0.327 ***	0.029	0.076	0.173

* In bold significant value *p* = 0.040.

**Table 3 healthcare-13-01866-t003:** Post hoc analysis of the relationship between uterine contraction location and pain (VAS2/VAS3) using Kruskal–Wallis with Bonferroni–corrected pairwise comparisons (*n* = 80).

		VAS2	VAS3
Uterine contraction	Infraumb./Supraumb.	23.73	34.31 *
	Infraumb./Umbilical	24.1 **	28.01 **
	Supraumb./Umbilical	1	1

* The value is significant at the 0.05 level; ** The value is significant at the < 0.001 level.

## Data Availability

The data presented in this study are available on request from the corresponding authors.

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
