# Peer review of "Beyond Pain Management: Skin-to-Skin Contact as a Humanization Strategy in Cesarean Delivery: A Randomized Controlled Trial"

_healthcare, 2025, doi:10.3390/healthcare13151866_

Round 1
Reviewer 1 Report
Comments and Suggestions for Authors
The study is valuable and well-structured, making an important contribution to the promotion of humanized care in the context of cesarean birth. The results are clinically relevant and may support changes in obstetric practice by integrating immediate skin-to-skin contact (SSC) into postoperative care protocols.
Suggestions for improvement:
-
Stylistic clarity: A minor language revision is recommended in some paragraphs to enhance fluency and ensure a more concise writing style.
-
Methodology: It would be helpful to provide a more detailed explanation of the sample size calculation rationale and the exact procedure for implementing SSC in the operating room.
-
Discussion: The discussion section could be strengthened by including a broader comparison with other similar RCTs and by highlighting the study's limitations (e.g., sample size, potential lack of blinding).
-
Implementation guidance: It would be useful to explicitly outline the necessary steps for implementing SSC in maternity units where this practice is not yet established, particularly in the context of emergency cesarean sections.
-
References: Consider adding an additional reference to European or national guidelines on multimodal analgesia after cesarean section to provide broader contextual grounding.
The manuscript is written in clear and generally fluent English. The scientific terminology is used appropriately, and the structure of the text is coherent. However, a minor language revision is recommended to improve stylistic clarity and fluency in certain sections, particularly in the methodology and discussion. This will enhance the overall readability and professional polish of the paper.
Reviewer 2 Report
Comments and Suggestions for Authors
This manuscript presents a randomized clinical trial evaluating the effects of immediate skin-to-skin contact (SSC) following caesarean section on postoperative pain, uterine contraction quality, and maternal satisfaction. The topic is highly relevant and timely, aligning with growing global interest in humanized obstetric care. However, while the findings are promising and clinically significant, there are several areas—both conceptual and technical—that require clarification or revision before publication.
1.) While the manuscript references a previous quasi-experimental study for calculating sample size, (reference of the study is needed!) the statistical rationale (e.g., assumptions about effect size, standard deviation) is not sufficiently detailed.
2.) There is a discrepancy in participant numbers: the abstract refers to 81 participants, but in the results section, it is not always clear whether analyses include 80 or 81 subjects. This needs clarification, particularly in Table 3 where "N=81" is noted despite one reported dropout. ((Page 6, Lines 227–229))
3.) The manuscript introduces several outcome measures, but some are inadequately defined or lack unit clarification. For example in Table 1, several quantitative variables such as age, weight, height, and BMI are presented with values in parentheses (e.g., 33.12 (3.57)), but the meaning of these parentheses—presumably standard deviations—is not clarified. Moreover, the lack of measurement units (e.g., years for age, kg for weight, cm for height, kg/m² for BMI) reduces the table’s clarity and scientific precision. These omissions should be corrected for accurate interpretation.
Additionally, the variable “Level of studies” appears here for the first time without prior explanation in the Methods section. There is no description of how the categories “Basic,” “Medium,” and “High” were defined or measured. This is problematic, especially since a significant difference (p < 0.001) was found between groups, making it a possible confounding factor. The lack of methodological transparency on this point needs to be addressed.
4.) Table 2 is confusing and internally inconsistent. While the heading states (N = 80), the subgroup sizes listed below add up to 81 (Control: n = 41, SSC: n = 40), contradicting the earlier mention of one participant dropout. This inconsistency must be corrected for clarity and data integrity.
Furthermore, the APGAR scores are included in the correlation matrix, yet no descriptive statistics or distributional data (e.g., means, SDs, or range) are provided for these variables in the results or any other table. Without these details, it is impossible to assess the clinical significance of the correlations presented or understand the variability in newborn condition across groups. This omission weakens the transparency of the findings and should be addressed.
5.) In Tables 2 and 3, some values are reported without specifying what statistical test was used or without indicating significance clearly in the table caption or footnotes. Use of asterisks (*) for significance should be consistently explained.
6.) Consider aligning the title more closely with journal conventions by avoiding abbreviations. Although the manuscript is generally well-written, some sentences—particularly in the Introduction and Discussion—are overly long or repetitive. For example, lines 66–69 repeat earlier statements and could be streamlined. Figure Interpretation: Figures 1–3 are relevant but not always clearly described in the main text. Adding brief interpretations of each in the results would help readers follow the findings more intuitively.
Round 2
Reviewer 2 Report
Comments and Suggestions for Authors
The manuscript is suitable for publication.